# Modulation of EndMT by Hydrogen Sulfide in the Prevention of Cardiovascular Fibrosis

**DOI:** 10.3390/antiox10060910

**Published:** 2021-06-03

**Authors:** Lara Testai, Vincenzo Brancaleone, Lorenzo Flori, Rosangela Montanaro, Vincenzo Calderone

**Affiliations:** 1Department of Pharmacy, University of Pisa, 56126 Pisa, Italy; lorenzo.flori@phd.unipi.it (L.F.); vincenzo.calderone@unipi.it (V.C.); 2Interdepartmental Center of Ageing, University of Pisa, 56126 Pisa, Italy; 3Department of Science, University of Basilicata, 85100 Potenza, Italy; vincenzo.brancaleone@unibas.it (V.B.); rosangela.montanaro@unibas.it (R.M.)

**Keywords:** gasotransmitter, endothelial mesenchymal transition, fibroblast phenotype, cardiovascular diseases, TGF-β

## Abstract

Endothelial mesenchymal transition (EndMT) has been described as a fundamental process during embryogenesis; however, it can occur also in adult age, underlying pathological events, including fibrosis. Indeed, during EndMT, the endothelial cells lose their specific markers, such as vascular endothelial cadherin (VE-cadherin), and acquire a mesenchymal phenotype, expressing specific products, such as α-smooth muscle actin (α-SMA) and type I collagen; moreover, the integrity of the endothelium is disrupted, and cells show a migratory, invasive and proliferative phenotype. Several stimuli can trigger this transition, but transforming growth factor (TGF-β1) is considered the most relevant. EndMT can proceed in a canonical smad-dependent or non-canonical smad-independent manner and ultimately regulate gene expression of pro-fibrotic machinery. These events lead to endothelial dysfunction and atherosclerosis at the vascular level as well as myocardial hypertrophy and fibrosis. Indeed, EndMT is the mechanism which promotes the progression of cardiovascular disorders following hypertension, diabetes, heart failure and also ageing. In this scenario, hydrogen sulfide (H_2_S) has been widely described for its preventive properties, but its role in EndMT is poorly investigated. This review is focused on the evaluation of the putative role of H_2_S in the EndMT process.

## 1. Introduction

Endothelial mesenchymal transition (EndMT) is a complex biological process in which endothelial cells lose their specific markers, such as vascular endothelial cadherin (VE-cadherin), and acquire a mesenchymal or myofibroblastic phenotype, expressing specific products, such as α-smooth muscle actin (α-SMA) and type I collagen [1]. From a histological point of view, during EndMT, the integrity of the endothelium is disrupted, manifesting a migratory, invasive and proliferative phenotype. Vascular endothelium can be viewed as a specialized form of epithelial tissue. Consistently, EndMT shares many molecular mechanisms with the best-known epithelial mesenchymal transition (EMT), a physiological reversible process, first described in the 1960s in chick embryos by Hay and today known to be required for normal embryonic patterning and organs’ formation [1]. 

Kovacic et al. first reported that EndMT represents a crucial step in the formation of septa and valves in mammalian hearts, during cardiogenesis; indeed, anomalous EndMT results in septal defects and valve abnormalities [2]. Of note, recent evidence suggests that this phenomenon also has an important role in various adult conditions, including fibrosis, wound repair and inflammation [2,3]. 

This theory has been formulated on the basis of the first pioneer studies, in which adult bovine endothelial cells with an endothelial-specific phenotype could differentiate into smooth muscle cells. The start of trans-differentiation was correlated with the disruption of cell–cell contacts and was induced by transforming growth factor (TGF-β1). This suggested that mature bovine large-vessel endothelial cells were able to acquire a smooth muscle phenotype [4].

However, the EndMT must be seen as a dynamic and reversible process, in which mesenchymal phenotype, under specific stimuli, can recover the endothelial feature. In this regard, Ubil and colleagues described a critical contribution of mesenchymal–endothelial transition (MEndT) in post-infarct cardiac neovascularization. They demonstrated that, in ex vivo and in vivo models, cardiac fibroblasts could adopt endothelial cell fate after stimulation of the p53 pathway, leading to angiogenesis and promoting the reparation of injured myocardium; vice versa, treatment with TGF-β repressed them [5,6]. 

## 2. Molecular Mechanisms Underlying the EndMT

A complex network of gene activation and repression programs are required for the initiation, execution and maintenance of switching phenotypes. In particular, EndMT occurs under several types of stimuli, including fibroblast growth factor (FGF), interferon gamma (IFN-γ) and metabolic factors, such as oxidized low-density lipoproteins (oxLDL), high-density lipoproteins (HDL) and high glucose (HG) levels. However, TGF-β is considered the prominent regulator of EndMT. Then, EndMT proceeds through several smad-dependent and smad-independent signaling pathways, including Notch, Wnt and PI3K/Akt pathways. These engage special transcription factors (snail, slug, twist and ZAB), which are responsible for repression of the endothelial phenotype as well as progression of the transition process, characterized by expression of mesenchymal markers [7,8,9,10,11].

### 2.1. TGF-β-Dependent EndMT

TGF-β and bone morphogenetic protein (BMP) belong to the TGF-β superfamily [12]. Analyzing the phenotypes of mice deficient in components of the TGF-β signaling cascade, the importance of the TGF-β signaling pathways in the spatial and temporal regulation of heart and blood vessel morphogenesis, as well as cardiovascular homeostasis, is evident. For example, TGF-β1-deficient mice show decreased age-associated myocardial fibrosis and improved cardiac compliance. Conversely, mice with TGF-β1 overexpression have significant ventricular fibrosis, with an increase of cardiac fibroblasts. Accordingly, treatment with NP-40208, an antagonist of the TGF-β receptor, attenuates myocardial fibrosis in mice [13]. Therefore, today, therapeutic interventions focused on normalizing perturbed TGF-β signaling are an emerging area of intense research.

Three isoforms of TGF-β have been described. TGF-β1, the most commonly studied in the context of pathological EndMT, intervenes, regulating the atrioventricular canal (AVC) in mice during morphogenesis [14]. TGF-β2 is the most important in developmental EndMT. For example, it seems to be involved in endocardial cushion formation, as observed by using TGF-β knockout embryos and cultured AVC explanted from TGF-β knockout mice [9,15,16,17,18]. Finally, the effects of the TGF-β3 isoform on EndMT are relatively unknown, during development as well as pathological conditions [19].

TGF-β is a dimeric cytokine produced from various cells in an inactive form, in which the amino-terminal part is non-covalently associated with the mature carboxy terminal peptide. Upon proteolytic cleavage, the bioactive TGF-β is released and can stimulate specific serine/threonine receptors (TGF- βRs) [20,21]. 

Although the exact stoichiometry in the signaling complex is not known, in response to interaction of TGF-βR type II with TGF-β, it dimerizes with the TGF-βR type I (also termed activin receptor-like kinase, ALK), phosphorylates and triggers the internalization of the TGF-β-receptor complex. EndMT may proceed via both canonical smad-dependent and non-canonical smad-independent pathways. The canonical smad-dependent pathway requires the phosphorylation of smad2 and 3 and their binding with the cytosolic smad4, to obtain a heterotrimer complex, that translocates into the nucleus, where it initiates the transition of endothelial cells. Antagonists of smad3 almost completely inhibit EndMT and fibrosis. Conversely, the activation of smad3 promotes fibrosis in a murine model [22,23,24]. Then, the transcription factors slug and snail repress endothelial markers (VE-cadherin and CD31), driving the cells towards the mesenchymal transition and fibrotic cardiovascular diseases. The TGF-β/smad/snail/slug pathway is central; indeed, an inhibition of snail/slug reduces EndMT as well as the occurrence of cardiovascular diseases [25].

TGF-β also activates the smad-independent signaling cascade, involving the engagement of several downstream kinases, such as mitogen-activated protein kinase (MAPK)/extracellular-signal-regulated kinase (ERK), phosphoinositide 3-kinase (PI3K), Rho-like GTPase and p38 MAPK [26,27]. Akt is also found downstream of the TGF-β signaling pathway and is activated in a RhoA-dependent manner. Furthermore, an inhibition of Akt reduces smad2 phosphorylation and transcription. Finally, GSK-3β inhibits pro-fibrotic TGF-β signaling via interaction with smad3 [28].

### 2.2. TGF-β-Independent EndMT

Several BMPs and their receptors (BMPRs) are also critical factors required for EndMT, and during embryogenesis they contribute to valve formation and cardiac septation [29,30]. The Notch pathway is also important during physiological cardiac development; indeed, mutations at this level are associated with defective valve formation [31], and the inhibition of Notch1 is associated with decreased EndMT and cardiac fibrosis [32].

Interestingly, Wnt signaling activation occurs upon cardiac injury and can start EndMT in the epicardium and contribute to generating cardiac fibrosis. In particular, β-catenin, the downstream effector, is stabilized and translocates into the nucleus, where it induces the transcription factor twist [33]. On the contrary, the disruption of Wnt signaling in epicardial cells decreases epicardial expansion as well as EMT and results in cardiac dysfunction and ventricular dilatation [34,35].

Inflammatory cytokines, such as tumor necrosis factor-α (TNF-α), interleukin-1β (IL-1β) and interleukin-6 (IL-6), may be triggers of TGF-β-driven EndMT, through the nuclear factor-κB (NF-kB) pathway [36]. Moreover, the vasoactive peptide endothelin-1 (ET-1) has been found to promote cardiac fibrosis and heart failure in diabetic hearts through stimulation of EndMT. In particular, studies on human endothelial cells have demonstrated that ET-1 is able to increase TGF-β-induced EndMT and that these effects involve the smad pathway [37,38].

Other cytokines may act as stimuli for EndMT, including IFN-γ, that acts through the Janus kinase (JAK)/signal transducer and activator of transcription (STAT) pathway, following the snail-dependent pathway [39]. FGF has been proposed as a gatekeeper of partial EndMT, since reduction of FGF signaling in mice represses TGF-β signaling [40]. 

PI3K/Akt signaling has also been found to be involved in fibrosis in mice. It is associated with an increased expression of MMP9 at the vascular level and recruits the transcription factor ZEB [41]. 

Illigens and colleagues showed that exogenous administration of vascular endothelial growth factor (VEGF) contributed to preservation of myocardial systolic and diastolic function, preventing apoptosis. Moreover, a reduced extracellular matrix (ECM) deposition and then anti-fibrotic effects, through negative modulation of the TGF-β/smad-dependent pathway, were reported [42].

Several studies have demonstrated that metabolic disorders, such as diabetes and dyslipidemia, are highly associated with the development of cardiovascular diseases, and EndMT is emerging as a further mechanism involved. In this regard, high levels of HDL show anti-fibrotic effects through the blockage of the TGF-β/smad/slug/ZEB signaling pathway; conversely, HG conditions have been demonstrated to induce EndMT by means of the Akt/PI3K/NF-kB pathway and are associated to fibrosis in diabetic patients [43,44]. Likewise, Tang et al. demonstrated endothelial cell transition in streptozotocin (STZ)-induced diabetes in mice and the involvement of the TGF-β/smad/snail pathway [45]. In this regard, Mishra et al. observed that AMP-activated protein kinase (AMPK), a key modulator of cardiometabolic homeostasis, inhibited TGF-β-induced smad3-dependent transcription. This hypothesis has been strengthened by using AMPKα knockout mice, in which fibrosis was promoted and cardiac function was compromised [46].

Promotion of EndMT may be considered as a further possible mechanism of Angiotensin II (AngII)-mediated endothelial dysfunction, since the treatment with irbesartan attenuated it and slowed atherosclerosis in type II diabetes mellitus patients [47].

Epigenetic modifications can also participate in EndMT and play important roles in cardiovascular conditions, where DNA methylation, histone modifications and RNA interference are recognized as the most involved. For example, microRNAs, such as miRNA-125b, Let-7c, Let-7g, miRNA-21, miRNA-30b and miRNA-195, seem to promote profibrotic processes and are at the basis of pathogenesis of atherosclerosis, cardiac fibrosis and heart failure. Overexpression of miRNA-126, miRNA-483 and miRNA-155 has been associated to abrogation of the TGF-β/smad pathway and to protective effects [48].

Furthermore, TGF-β1 treatment in human coronary endothelial cells resulted in aberrant promoter methylation and thereby reduced gene expression of RASAL1 (RAS protein activator-like 1), which contributed to EndMT. Indeed, an aberrant promoter methylation of RASAL1 was also observed in an experimental murine model of cardiac fibrosis and in end-stage heart failure patients [48].

Interactions amongst these pathways increase the complexity of the process, since they all converge and induce the expression of transcription factors, such as snail, slug and twist, and ultimately modulate EndMT (Figure 1).

## 3. EndMT: Implications in Cardiovascular Fibrosis

Fibrosis is a complex process, in which tissue repair after damage becomes excessive and out of control, resulting in excessive formation of fibrous connective tissue. It can affect any tissue and cause organ dysfunction in different pathologies, such as heart disease, interstitial lung disease, liver cirrhosis, progressive systemic sclerosis and diabetic nephropathy [49].

Deposition of fibrotic scar tissue in the heart is typical after myocardial infarction in the post-ischemic stage, and in vessels of hypertensive subjects, it represents the way through which atherosclerosis develops. Other conditions can also induce and promote fibrotic processes in heart tissue, such as hypertensive cardiomyopathy, diabetic hypertrophic cardiomyopathy, idiopathic dilated cardiomyopathy and also physiological ageing [50,51].

### 3.1. EndMT and Vascular Fibrosis

Vascular fibrosis is associated with many diseases and their pathological progression, including atherosclerosis, which is one of the primary causes of the development of cardiovascular diseases. Indeed, atherosclerosis is characterized by accumulation of ECM proteins, primarily collagen and fibronectin, in the vascular media, contributing to structural vascular remodeling, proliferation of vascular smooth muscle cells and inhibition of matrix degradation, and is responsible for the thickness of the wall of arteries and formation of plaques [52]. Endothelial dysfunction is a critical event in the development of atherosclerosis; indeed, in physiological conditions, the endothelium is a monolayer squamous epithelium located in the luminal surface of the blood vessels and is a major regulator of vascular homeostasis. In particular, by the production of nitric oxide (NO), it preserves structural and functional integrity of the vessels. Reactive oxygen species (ROS) production is a major atherosclerotic risk factor that promotes NF-kB signaling and then inflammation. Proinflammatory cytokines contribute to increase the atherosclerotic region, through arterial plaque formation and additional cell apoptosis, leading to lipids’ expulsion into adjacent plaque areas [53]. 

Interestingly, plaque formation during atherosclerosis has been associated with the accumulation of mesenchymal cells in the arterial intima, deriving from transition of endothelial cells. It is noteworthy that, in plaques of murine atherogenic models and in humans, endothelial and mesenchymal markers have been found, demonstrating the presence of a transitioning state of EndMT. Recently, it has been suggested that mesenchymal cells advance the progression of atherosclerosis, since they secrete proinflammatory molecules and synthetize ECM proteins and metalloproteases (MMPs), which facilitate plaque build-up and regulate plaque stability [54]. Interestingly, in endothelial cells, TGF-β predominantly binds ALK-1, promoting smad1 and smad5, and contributes to regulation of vascular homeostasis, cell proliferation and angiogenesis; in contrast, when TGF-β binds ALK5, it induces the activation of smad2 and smad3, which inhibit cell proliferation and facilitate EndMT [55,56]. Besides canonical TGF-β signaling, TGF-β can induce EndMT in a non-canonical way, through the activation of kinase-driven signaling pathways and the downstream induction of the mesenchymal transcription factors [55].

Angiogenesis is a vascular remodeling process, promoted by VEGF and other factors that stimulate proliferation and migration of vascular endothelial cells to form new blood vessels. As previously described, TGF-β, beyond its role in physiological [57] and degenerative processes, has been recognized as one of the key factors inducing the expression of VEGF in endothelial cells via the ALK-5 pathway, thus creating a pro-angiogenic milieu for the tumor or for the EndMT [58]. Indeed, excessive sprouting is associated with the TGF/VEGF pathway. However, FGF signaling is involved in angiogenesis modulation and in the maintenance of vascular integrity and endothelial function. Crosstalk between FGF and TGF-β exists. In this regard, recent studies show that FGF inhibition at the endothelial level is associated with TGF-β activation [57].

Very interestingly, EndMT of the specialized brain endothelial cells, forming the blood–brain barrier, has been described in different conditions of neuroinflammation and neurodegeneration, typical of central nervous system pathologies, such as multiple sclerosis [59].

### 3.2. EndMT and Cardiac Diseases

Cardiac fibrosis is a hallmark of the heart’s pathological remodeling response to mechanical or biochemical stress. This process serves to maintain the structural integrity of the myocardium, but inevitably leads to reduced contractile capacity. Indeed, cardiac fibrosis is characterized by increased stiffness of the heart valves, due to excessive proliferation of cardiac fibroblasts, accumulation of myofibroblasts and to deposition of ECM in the cardiac muscle. In particular, the ECM is a highly durable and mechanically stable fiber-containing structure that serves as a scaffold for cells and can be useful for wound healing and tissue regeneration. Nevertheless, unlike cardiomyocytes, cardiac fibroblasts are unable to generate an action potential and are not excitable (although they are able to electrically couple with each other and with neighboring cardiomyocytes), causing impaired mechanical–electrical coupling and arrhythmias [60]. On the other hand, impaired tissue function, myocardial dysfunction and ultimately heart failure are also observed [61]. 

Myocardial infarction is the most common cause of cardiac fibrosis. However, various other conditions, such as hypertension, diabetic hypertrophic cardiomyopathy, idiopathic dilated cardiomyopathy and ageing, may be responsible for reparative mechanisms, addressed to the replacement of dead cardiomyocytes with a collagen-based scar. 

Another pathological condition promoting myocardial fibrosis is diabetic cardiomyopathy (DCM); interestingly, about 75% of patients with unexplained idiopathic dilated cardiomyopathy are found to be diabetic [62]. Endoplasmic reticulum stress, mitochondrial dysfunction, sympathetic nervous system activation, excessive oxidative stress, increased inflammation and abnormal coronary microcirculation are characteristic symptoms. These pathophysiological changes result in fibrosis, hypertrophy, diastolic/systolic dysfunction and ultimately systolic heart failure. In particular, cardiac interstitial fibrosis is a major feature of DCM [63], which includes the overproduction and deposition of myocardial interstitial collagen, resulting in myocardial stiffness and cardiac dysfunction. 

Another aspect that can greatly affect the cardiovascular system is ageing. Ageing causes a decreased production of NO and a process of generalized endothelial dysfunction at the vascular level. At the cardiac level, consequently, with the progress of ageing, compensatory mechanisms addressed at hypertrophy [64,65,66,67,68] cause an increase in the left ventricular wall, which in the long term worsens the overall cardiac performance, and also triggers fibrotic processes as a final manifestation. Along with the mechanical consequences, cardiac fibrosis can also slow down the propagation of the electrical impulse and affect heart rate, suggesting that ageing can also promote cardiac arrhythmias [53].

First, in 2007, Ziesberg et al. hypothesized that EndMT is a process at the basis of cardiac fibrosis. Indeed, they observed that under stimulation, adult human coronary endothelial cells trans-differentiated into fibroblasts, and bone marrow cells could contribute to the cardiac fibroblast population. Moreover, they hypothesized that TGF-β1/smad was the signaling pathway through which EndMT developed. In fact, the treatment with rhBMP-7, a TGF-β antagonist, was able to preserve the endothelial phenotype and ameliorate the progression of fibrosis in cells as well as in a murine model of heart disease [9], demonstrating that it was responsible for the total pool of cardiac fibroblasts. According to this evidence, Goumans’ group confirmed the key role of the growth factor in EndMT in different in vitro and in vivo models [69].

More recently, Kong et al. [70] reported that accumulation and production of collagen and cardiac fibroblasts in human pathological patients were related to the process of EndMT.

Kovacic and colleagues also demonstrated that EndMT plays a main role in cardiac fibrosis that progresses to heart failure under hypertrophic cardiomyopathy, diabetes-induced cardiac fibrosis and genetic deficiency of PAI-1 in aged mice [2].

This is a relevant aspect, since a significant number of non-myocytes are present in the heart. In particular, it has been reported that about 64% of the non-myocyte cell population in the mouse and 54% in the human heart are endothelial cells [71].

## 4. Biosynthesis, Metabolism and Sulfhydration of Cellular Targets by H_2_S

Hydrogen sulfide (H_2_S) is a gaseous mediator that is endogenously generated and exerts multiple effects. In particular, H_2_S can be synthesized through enzymatic pathways, involving cystathionine β-synthase (CBS), cystathionine γ-lyase (CSE), cysteine aminotransferase (CAT) and 3-mercaptopyruvate sulfur transferase (3-MST) [72,73,74,75,76,77] (Figure 2). 

All the enzymatic paths involve L-cysteine as the main substrate to be processed and are pirydoxal-5′-phosphate-dependent. Interestingly, biosynthesis of H_2_S leads to other by-products, such as pyruvate or serine, respectively entering cell energetic metabolism or modulating vascular function through NO release [78,79]. All H_2_S-generating enzymes are widely distributed within the body with different relative expressions. In particular, CBS is the most abundant in the central nervous system, though it has been reported in different peripheral tissues, including vasculature, liver and kidney [74,80,81]. In addition, CSE is the major enzyme producing H_2_S in the cardiovascular system and it has been found particularly in endothelial as well as smooth muscle cells, as well as in myocardiocytes [73,82,83], where CAT and 3-MST have also been described [76,84]. 

H_2_S generated within the cells is free to interact with intracellular targets or diffuse outside the cell to modulate other cell functions. Several targets have been associated with H_2_S action, including ion channels, such as ATP-sensitive potassium channels (K_ATP_), voltage-gated potassium channels (Kv7), transient receptor potential channel (TRPV) or L/T-type Ca^2+^ channels, enzymes such as phosphodiesterase, phospholipase A2, PI3K and Akt, or transcription factors such as STAT-3, nuclear factor erythroid 2-like 2 (Nrf-2) and NF-kB [82,85,86,87,88,89]. 

H_2_S is mainly known to regulate vascular function by inducing vasodilation, through inhibition of phosphodiesterase and activation of K_ATP_ and Kv7 channels; consistently, deletion of CSE leads to hypertension [90]. Nevertheless, H_2_S regulates multiple functions within the body. In particular, it exerts anti-inflammatory properties by reducing pro-inflammatory mediators and activating resolution pathways [91,92,93,94]. 

Furthermore, H_2_S has cytoprotective effects not only due to its scavenging properties, but also to other signaling pathways associated with an increase in glutathione (GSH) levels, reduction in Ca^2+^ influx and activation of mitochondrial ATP-sensitive potassium channels (mitoK_ATP_) [87,95,96,97].

The interaction mode between H_2_S and its molecular targets depends on the target itself and, frequently, it occurs through a process known as protein sulfhydration [98,99] (Figure 2). This mechanism consists of a chemical reaction on cysteine residues, generating a -S-SH moiety forming chemical species referred to as bound sulfane-sulfur. These species can modulate the protein activity and, at the same time, can store H_2_S molecules that can be later released for biological actions. In the last decade, several examples of sulfhydration in the control of biological processes regulated by H_2_S have been reported. 

For instance, Altaany and co-workers showed that endothelial nitric oxide synthase (eNOS) activity is enhanced following sulfhydration through the enhancement of eNOS phosphorylation [100]. Another example of sulfhydration reaction has been reported for caspase 3, whose activity is suppressed, preventing neuronal injury associated with ischemia/reperfusion (I/R) events [101]. Conversely, activity of NF-kB subunit p65 is enhanced following sulfhydration, resulting in suppression of the apoptotic pathway in macrophages following TNFα stimulus [102]. In addition, H_2_S can also drive sulfhydration of peroxisome proliferator-activated receptor-γ (PPARγ) at Cys 139, increasing its nuclear translocation to DNA, thus promoting expression of genes associated with adipogenesis [103]. This mechanism could explain the switch from glucose to triglycerides storage induced by H_2_S. 

Finally, H_2_S has been observed to modulate, via sulfhydration, Keap1-Nrf-2 and promote the gene transcription of antioxidant agents. Recent evidence shows that H_2_S can also increase Akt (Ser473) phosphorylation levels to improve damage in disease states [104].

Once H_2_S has exerted its effects, although it is relatively stable in blood and intracellular/interstitial fluids, it enters a catabolic pathway, where it serves as a scavenger of free radicals or hemoglobin. This feature is crucial, since H_2_S represents a relevant antioxidant molecule also involved in maintaining GSH/glutathione disulfide (GSSG) balance. Furthermore, H_2_S can be methylated or oxidized, thus generating multiple by-products, including sulfate, sulfide or thiosulfate, all of which are excreted via the urinary tract [105]. H_2_S can also be exhaled in breath, as demonstrated in animals and human volunteers following intravenous administration of NaHS [106,107]. However, despite the massive amounts of studies describing H_2_S biosynthesis pathways, little is known about its metabolism/elimination, apart from the chemical/biochemical reactions described above (Figure 2).

## 5. Contribution of Hydrogen Sulfide in Cardiovascular Fibrosis Associated with EndMT

### 5.1. Vascular Fibrosis and Hydrogen Sulfide

H_2_S is an important endogenous mediator with a key role in the cardiovascular system. Several studies have demonstrated a possible connection with TGF-β, and therefore its benefits on vascular fibrosis. In this regard, it is noteworthy to underline that Mallat and co-workers have confirmed that inhibition of TGF-β signaling could promote the development of atherosclerosis with decreased collagen content in ApoE^−/−^ mice [108]. This also occurs in uremia-accelerated atherosclerosis (UAAS), a common condition in patients with diabetic nephropathy. Indeed, the CSE/H_2_S system displays a protective role in UAAS. Lu and colleagues have demonstrated in a recent study that the use of NaHS, a hydrophilic fast H_2_S release, could suppress the formation of atherosclerosis via the degradation of TGF-β, as well as the reduction of smad3 phosphorylation. Therefore, the H_2_S pathway regulates TGF-β/smad3 signaling in UAAS mice, suggesting that the TGF-β/smad3 axis is responsible for CSE/H_2_S-dependent vascular protection. Moreover, such effect is specific for H_2_S, since CSE inhibitors, such as propargylglycine, can conversely accelerate the atherosclerosis development [109]. This evidence highlights that the control operated by H_2_S over TGF-β signaling, crucial for EndMT, could slow down the whole fibrotic process. 

Interestingly, CSE-knockout mice fed an atherogenic diet exhibited severe atherosclerosis, suggesting that the CSE/H_2_S pathway is somehow crucial to limit atherosclerosis development [110,111], which is also dependent on activation of inflammatory macrophages [112]. 

However, the mechanisms underlying the effects of H_2_S in the suppression of fibrosis are different and involve diverse pathways, and thus have to be found in its pleiotropic actions [113]. For instance, vascular fibrosis can develop following chronic hypertension and vasorelaxant effects driven by the H_2_S pathway control fibrotic process by reducing blood pressure, through the modulation of channels’ activity, cyclic guanosine monophosphate-protein kinase G (cGMP-PKG) pathways [114,115] and inhibition of angiotensin-converting enzyme (ACE) [116]. Although vasorelaxation can induce beneficial action over fibrosis per se, the inhibitory activity of H_2_S on ACE also abrogates vascular fibrosis associated with AngII proliferative pathways. Another alternative mechanism involving H_2_S signaling in the control of vascular fibrosis is associated with its anti-inflammatory properties, since H_2_S shows beneficial effects in cardiovascular remodeling through suppression of CD11b-positive leukocytes’ migration [117]. Furthermore, the reduction of oxidative stress in endothelial cells operated by H_2_S can also contribute to its antifibrotic properties [118]. It is well-known that the activation of the eNOS/NO pathway is a crucial event in the control of vascular homeostasis, and recent findings report that H_2_S shows an interplay with eNOS activation [78]. Indeed, H_2_S shows protective effects in hypertensive cardiovascular disease and this effect is mediated by the eNOS/NO axis [79,119]. 

In addition, the endogenous CSE/H_2_S system has been shown to protect against the formation of UAAS via an alternative pathway involving the activation of protein kinase CbII (CPKCbII)/Akt signaling [120]. Thus, it should not be surprising that H_2_S biosynthesis as well as CSE expression drop down in conditions of vascular calcification [121] or following the development of neointimal hyperplasia associated with induction of balloon injury. Again, this effect is reversed by H_2_S treatment [122].

A typical risk factor for vascular fibrosis is hyperhomocysteinemia (HHcy), together with hypertension, hyperglycemia and dyslipidemia [52]. HHcy is characterized by high tissue and plasma homocysteine (Hcy) levels and it is mainly due to impaired renal clearance and malnutrition [123]. HHcy leads to vascular fibrosis, causing several disease conditions, including peripheral and cerebrovascular coronary occlusion as well as venous thromboembolism [124,125,126]. The link between HHcy and vascular fibrosis is found upon an excessive ECM deposition, together with unbalanced elastin/collagen proportions occurring through an increase of intracellular calcium release [127]. The profibrotic effect caused by Hcy is due to activation of MMPs, involved in the atherosclerotic plaque formation, leading to the alteration of ECM metabolism and the promotion of collagen deposition [128]. The interplay between Hcy and H_2_S has been shown by Sen and colleagues as they demonstrated that activation of both MMP-2 and MMP-9 by HHcy was controlled following H_2_S supplementation [129]. Of course, this is not the only study reporting the link between H_2_S and Hcy, as both are part of a complex network of biochemical reactions referred to as the trans-sulfhydration pathway. Indeed, several reports show that high levels of Hcy can inhibit CSE activity [130]. Thus, in a pathological condition, HHcy can alter the whole trans-sulfhydration pathway through this mechanism [131]. Therefore, such a condition results in a drop of endogenous levels of H_2_S due to a suppressed biosynthesis within the body. The accumulating evidence indicating H_2_S as a crucial controller of vascular function leads to the straightforward conclusion that a lack of it results in pathological conditions affecting vascular tissues, including hypertension and fibrosis. Thus, it is not surprising that SG1002, a donor of H_2_S, is able to normalize the histological and molecular scores for hypertrophy and fibrosis, leading to a decrease in collagen accumulation in HHcy mice and, consequently, preventing vascular fibrosis [123].

Interestingly, a key point is also represented by the association between the fibrotic process, hypertension and AngII. As mentioned earlier, hypertension is a condition resulting from vascular fibrosis and might also be linked to Hcy levels. Indeed, Hcy induced the activation of MMP-9 together with the synthesis of collagen in endothelial cells, and this mechanism is driven by AT_1_-receptor activation by AngII [132]. Therefore, when an HHcy condition occurs, a reduction in H_2_S biosynthesis is observed, and this can undermine its ability to inhibit ACE, resulting in increased AngII-dependent profibrotic stimulation [133]. This aspect has also been investigated in pulmonary vasculature, where administration of NaHS is able to decrease the mean pulmonary arterial pressure and to inhibit smooth muscle cells’ proliferation in the pulmonary artery wall [134]. In addition, administration of exogenous H_2_S also decreases the expression of collagen I and III in the pulmonary arteries, again suggesting that H_2_S plays an important role in the development of the fibrotic process and vascular structural remodeling, thus controlling homeostasis in pulmonary arterial pressure [134].

It is intuitive that the use of therapeutic tools aiming to rescue physiological H_2_S levels could represent a possible way to tackle the fibrotic process as well as other vascular-based impairments such as hypertension at different levels (Figure 3). Indeed, H_2_S could modulate activation of the EndMT process as well as interfere with intermediate steps towards fibrosis.

### 5.2. Cardiac Fibrosis and Hydrogen Sulfide

Necrosis of cardiomyocytes in ischemic hearts triggers a strong inflammatory response and promotes interstitial and perivascular fibrosis due to biochemical, geometric and biomechanical changes of the ventricular wall not affected by the lesion [135]. The high production of pro-inflammatory cytokines (TNF-α, IL-1β and IL-6) accompanies myocardial damage and hypertrophic tissue remodeling, contributing to the deposition of fibrotic tissue.

Interestingly, a reduction of H_2_S levels and of CSE expression has been found in post-ischemic hearts [136]. Furthermore, H_2_S levels have been found to be inversely correlated with the severity of coronary heart disease [137].

Indeed, H_2_S shows cardioprotective activity against I/R injury and it is considered an important modulator of the ischemic preconditioning process. The mechanisms of action responsible for the effects of H_2_S are heterogeneous, for instance, protein sulfhydration of mitoK_ATP_ and mitochondrial voltage-gated potassium channels (mitoKv7). It is noteworthy that Kv7.4 channels have been recently described in cardiac mitochondria, where they seem to play a cardioprotective role, in inhibition of mast cell degranulation and inflammatory processes, as well as in antioxidant and pro-angiogenic action [96,138]. H_2_S seems to also be involved in suppression of pro-fibrotic mechanisms associated with myocardial ischemia through the involvement of different signaling pathways, such as Nrf2 [139,140,141], signaling pathways involving miRNAs [136,142,143] and mitochondrial protection mechanisms [104,144,145,146,147]. 

Experimental evidence shows that NaHS and slow-release H_2_S donors such as AP39, diallyl disulfide (DADS) and GYY4137 are able to reduce oxidative stress and apoptotic mechanisms [148,149,150], mitochondrial permeability transition pore (mPTP) opening [144,145,146], inflammatory responses and cardiomyocyte death [151] and iNOS expression in experimental models of myocardial infarction [152]. NaHS is also able to increase heme oxygenase-1 (HO-1) expression [152], to promote pathways of GSK-3β/β-catenin [153], cGMP-dependent PKG/phospholamban [154] and to promote angiogenesis [155] and autophagy in elderly hearts [156,157].

GYY4137 preserves cardiac function, attenuates adverse remodeling and can exert post-ischemic cardioprotective effects. Indeed, hearts treated with GYY4137 had left ventricular fibrosis significantly lower than untreated hearts after myocardial infarction. Greater blood vessel density was found in the left ventricular scar area of the GYY4137-treated animals compared to all other infarcted groups. Despite preserved left ventricular structure and function, treatment with GYY4137 increased the levels of atrial natriuretic peptide (ANP) and brain natriuretic peptide (BNP) in association with increased cGMP levels, parallel to higher cGMP-type I protein kinase (cGKI) levels [158].

Daily intraperitoneal administration of GYY4137 for 4 weeks to spontaneously hypertensive rats (SHR) decreased systolic blood pressure and inhibited myocardial fibrosis. This kind of treatment reduced the collagen deposition in the left ventricle, the ratio of perivascular collagen area vs. lumen area in perivascular regions and the concentration of hydroxyproline, collagen I and III mRNA expression and cross-linked collagen. GYY4137 also inhibited AngII-induced neonatal rat cardiac fibroblast proliferation, reduced the number of S-phase fibroblasts, decreased the expression and protein synthesis of collagen I and III mRNA, attenuated oxidative stress and suppressed α-SMA, by modulating the expression of TGF-*β*1 and the phosphorylation of smad2. These results showed that GYY4137 improved myocardial fibrosis, possibly through a mechanism involving inhibition of oxidative stress, blockade of the TGF-*β*1/smad2 signaling pathway and decreased expression of α-SMA in cardiac fibroblasts [150].

Even more innovative H_2_S donors, such as ADT-OH (H_2_S-aspirin hybrid molecule) or ZYZ-802 (a cysteine derivative), were able to positively intervene in the ischemic lesion by activating the AMPK signaling pathway [146] and reducing the miRNA-30 family [136].

Polhemus and co-workers demonstrated that administration of diallyl trisulfide (DATS), a long-acting H_2_S-donor organic polysulfide compound present in garlic, could attenuate left ventricular dilatation and dysfunction in a model of pressure-overload heart failure, attenuating the development of perivascular and intermuscular fibrosis and then cardiac hypertrophy [159].

Finally, it is interesting that SG-1002, an orally active H_2_S donor, is able to positively act in the fibrotic damage following a heart failure condition and in general of myocardial dysfunction, favoring the adiponectin-AMPK signaling pathway [160] and increasing the bioavailability of NO [161].

The metabolic dysregulation characteristic of diabetes, including hyperglycemia, hyperlipidemia and oxidative stress, causes the death of cardiomyocytic cells. The early stages of cardiac remodeling following diabetes are generally asymptomatic, with myocardial changes and damage almost exclusively at the molecular level. In the middle phase of remodeling, progressive hypertrophy of cardiomyocytes and myocardial fibrosis result in a reduced ejection fraction [162]. 

In patients with diabetes, as well as in rats treated with STZ, downregulation of the gene expression of the enzymes involved in the endogenous biosynthesis of H_2_S, with consequent reduction of circulating levels of H_2_S, was frequently found [163,164,165]. Growing evidence suggests that H_2_S exogenous administration could exert cardioprotective effects in these pathological conditions. There is evidence that the JAK/STAT signaling pathway participates in the protective effects of exogenous H_2_S against myocardial fibrosis in diabetes mellitus, though other studies point to an involvement of the H_2_S-forkhead box protein O1 (FoxO1) pathway in the pathogenesis of diabetic cardiomyopathy. In this regard, in a rat diabetic model induced by STZ administration, a condition of myocardial fibrosis was related to a significant decrease in the expression of tissue inhibitor of metalloproteinase-1 (TIMP1) and MMP2 and an increase in the expression of MMP7, MMP11, MMP13 and MMP16. These changes were significantly reversed after treatment with NaHS, improving myocardial hypertrophy and collagen deposition. This also happened through the engagement of other types of cellular mediators, including the reduction of the expression of Grp78, caspase-12 and CHOP [166]. This evidence suggests that H_2_S could attenuate diabetes-induced cardiac fibrosis through the modulation of MMP/TIMP expression, the regulation of TGF-β1 and other possible intracellular pathways, including the regulation of the PKC-ERK1/2/MAPK signaling pathway and the inhibition of inflammatory reactions [167,168,169].

Subsequent studies confirmed the protective effect of H_2_S against diabetes-induced myocardial fibrosis and demonstrated that it was also associated with the attenuation of autophagy through upregulation of the PI3K/Akt1 signaling pathway [166]. 

In 2018, Liu et al. found that the myocardial cell matrix was markedly disordered in diabetic rats; in fact, both myocardial interstitial fibrosis and collagen III deposition were increased. The expression of TGF-β, eIF 2α, GRP94, caspase-3, TNF-α, NF-κB, MDA and 4-HNE were significantly increased, the expression of JAK-1/2 and STAT1/3/5/6 were upregulated, while CSE, SOD, GSH and Bcl-2 were downregulated. Compared to the diabetic group, these changes were reversed in the H_2_S group [164]. 

Likewise, GYY4137 improved HG-induced oxidative stress and apoptosis in hearts. In particular, H_2_S induced FoxO1 phosphorylation and nuclear exclusion with an Akt-independent mechanism, supporting the idea that H_2_S can also inhibit dilated cardiomyopathy progression through this transcription factor [170].

Finally, circulating and cardiac H_2_S levels were analyzed in a mouse model of high-fat diet-induced cardiomyopathy (HFD). HFD feeding for 24 weeks contributed to reduce both circulating and cardiac H_2_S and induced hallmarks of type 2 diabetes. Marked cardiac dysfunction, cardiac hypertrophy and fibrosis were also observed. SG-1002 restored physiological levels of the gasotransmitter, improved some of the metabolic disorder markers and reduced diet-induced cardiac dysfunction. Further analysis revealed that H_2_S therapy restored adiponectin levels and suppressed cardiac stress typical of HFD feeding [160].

Mice deficient in the H_2_S-producing enzyme, cystathionine γ-lyase (CSE KO), showed reduced cardiac mitochondrial content compared to wild-type hearts. In contrast, mice overexpressing CSE (CSE Tg) and mice supplemented with SG-1002 showed improved cardiac mitochondrial content. Further analysis revealed that cardiac H_2_S levels affected nuclear localization and transcriptional activity of proliferator-activated peroxisome receptor γ coactivator 1α (PGC1α). Studies aimed at evaluating the underlying mechanisms found that H_2_S required AMPK to induce PGC1α signaling and mitochondrial biogenesis. Restoring H_2_S levels with SG-1002 in the context of heart failure increased cardiac mitochondrial content and improved mitochondrial respiration, the efficiency of ATP production and heart function [171].

In 2017, Liang et al. investigated the role of H_2_S in myocardial fibrosis induced by chronic alcohol intake. A mouse model of cardiomyopathy was induced by administration of 4% ethanol solution in drinking water for 12 weeks. In this experimental condition, NaHS showed positive effects on myocardial fibrosis and deposition of collagen [172].

In 2018, Liu et al. investigated the role and regulatory mechanism of H_2_S in the improvement of thyroxine-induced rat myocardial fibrosis through autophagy interference by regulation of the PI3K/AKT1 signaling pathway activity and the expression of related miRNA. In the thyroxine group, myocardial fibrosis was more significant, protein expression of PI3K/AKT and autophagy-related proteins were significantly decreased, as well as miR-221 expression if compared to the control group, while miR-21, miR-34a and miR-214 expressions were significantly higher. Conversely, all of the changes mentioned above were reversed with H_2_S treatment, which demonstrated the positive function of this gasotransmitter in ameliorating thyroxine-induced myocardial fibrosis in rats. The mechanism of this enhancement may be correlated with autophagy triggered by the upregulation of the expression of the PI3K/AKT signaling pathway and by the downregulation of miR-21, miR-34a and miR214 expressions [173] (Figure 4).

## 6. Role of Hydrogen Sulfide in EndMT

A growing body of evidence shows that H_2_S plays a relevant role in the modulation of EMT, in several types of fibrosis. In this regard, Fang and colleagues demonstrated that a deficient endogenous CSE/H_2_S system was at the basis of the development of bleomycin-induced pulmonary fibrosis in rats. Exogenous application of NaHS interfered with lung fibrosis pathogenesis, through reduction of oxidative stress and suppression of migration, proliferation and myofibroblast trans-differentiation in human lung fibroblast cells. This evidence suggested that H_2_S could be an important regulator of pathogenesis of pulmonary fibrosis [174]. Later, they observed that H_2_S attenuated fibrosis induced by treatment with TGF-β1 in alveolar epithelial cells and by reduction of smad2/3 phosphorylation, but independently from activation of K_ATP_, a well-known target of the gasotransmitter involved in cell protection [175].

A more recent study indicates that exogenous H_2_S inhibited paraquat-induced EMT in human alveolar epithelial cells through regulating the TGF-β1/smad2/3 signaling pathway, also providing a novel idea for the clinical treatment of poisoning [176]. 

Indeed, the anti-fibrotic effects of H_2_S have been widely investigated in airways, to evaluate its contribution in the maintenance of respiratory function following the exposition to environmental toxins. In this context, GYY-4137 inhibited TGF-β1-induced cell morphological changes and EMT in human bronchial epithelial cells [177]. Nickel compounds are known to be common environmental and occupational carcinogens. In cell models, nickel is able to upregulate the protein levels of TGF-β1 and of smad2/3 phosphorylation, triggering the EMT process. Conversely, exogenous NaHS alleviates the nickel-induced EMT and the migration ability of lung cancer cells, indicating that NaHS might also have protective effects against nickel-induced lung cancer progression [60].

EMT has also been described as a key regulator of cancer progression and metastasis, and in this regard, Wang and colleagues demonstrated that H_2_S could modulate angiogenesis and the progression of non-small cell lung cancer via HIF-1α activation and stimulation of VEGF expression [60]. EMT has also been described during peritoneal fibrosis, to explain the loss of mesothelial cells and the occurrence of myofibroblasts. It is a characteristic condition described as a major cause of ultrafiltration failure of peritoneal dialysis. In fact, under HG stimulus, mesothelial cells undergo EMT, leading to a loss of epithelial tight junctions and an increase of the myofibroblast marker, α-SMA. In that experimental model, rats treated with glucose plus lipopolysaccharide for 28 days produced peritoneal fibrosis, and of note, NaHS reduced the deposition of collagen in the sub-mesothelial zone and inhibited inflammatory markers and the TGF-β1/smad signaling pathway [178].

In agreement with the previous evidence, H_2_S counteracted Ang II-induced EMT in renal tubular epithelial cells, through mechanisms involving direct and indirect inactivation of TGF-β1. This is the first work, to our knowledge, in which a possible mechanistic insight into the anti-fibrotic actions of H_2_S has been suggested. The authors supposed that beyond indirect antioxidant action, H_2_S could affect AngII-induced EMT through direct effects on the structural modifications at the level of an inactivated form of TGF-β1. Indeed, the cleavage of the disulfide bond in the active dimeric TGF-β1 by H_2_S could represent a key modification of Cys residues [60]. 

Despite the body of evidence collected on the putative role of H_2_S in the modulation of EMT, only recently has the first study been published, revealing that H_2_S can attenuate EndMT induced by ER stress at the cardiovascular level. In particular, Ying and colleagues hypothesized that H_2_S is able to suppress the EndMT in human umbilical endothelial cells submitted to endoplasmic reticulum stress. Under stimuli, EndMT is triggered, upregulating the phosphorylation of smad2, independent of the TGF-β pathway. The Src pathway is also involved in EndMT and is activated during ER stress. It is noteworthy that the treatment with H_2_S reverted EndMT, and conversely, the inhibition of Src kinase, by specific antagonists or silencing, abolished the protective effects of H_2_S [179]. 

More recently, a preventive role of H_2_S in the vascular remodeling of pulmonary arterial hypertension has been hypothesized. Indeed, EndMT has been suggested as a process involved in fibrosis in patients with pulmonary artery hypertension and in several animal models. In human pulmonary artery endothelial cells, transfection with CSE plasmid or exogenous H_2_S significantly repressed TGF-β1-induced expression of the mesenchymal markers and upregulated the expression of the endothelial markers, accompanied by the suppression of the NF-κB/Snail pathway. On the contrary, pretreatment with an inhibitor of CSE reversed it, revealing another possible mechanism through which H_2_S may play a preventive role towards pulmonary hypertension. Furthermore, the authors speculated that H_2_S could regulate the epithelial/endothelial phenotypic transition by sulfhydrating NF-kB, or via Snail at the level of cysteine residues [180].

Although not directly related to the EndMT, several studies show anti-fibrotic effects of H_2_S and its ability to regulate underlying pathways. For example, GYY4137 improved myocardial fibrosis by a mechanism involving the inhibition of the TGF-β1/smad2 signaling pathway and a decrease in α-SMA expression in cardiac fibroblasts [150]. This pathway is implicated in the reduction of myocardial fibrosis promoted by a liposomal formulation of S-propargyl-cysteine and endowed with cardio-protection in in vitro and in vivo models of I/R and in heart failure, and with angiogenetic effects. Importantly, in an in vivo model of heart failure, liposomal ZYZ-802 markedly inhibited myocardial fibrosis via the inhibition of the TGF-β1/smad signaling pathway [181]. However, in rat diabetic myocardial fibrosis, H_2_S had preventive effects through negative regulation of the Wnt-pathway and downregulation of TGF-β/smad3 signaling [169]. It is well-known that the CSE-H_2_S pathway shows antiatherogenic effects and it may reduce the size of atherosclerotic lesions at the vascular level. On the other hand, the role of EndMT in the progression of atherosclerosis is also recognized. It is noteworthy that a common molecular mechanism has been identified for H_2_S and EndMT, showing that in Apo^−/−^ mice treated with NaHS, TGF-β protein expression and smad3 phosphorylation decreased. Conversely, the inhibition of the CSE-H_2_S pathway reverted it. Therefore, at least in part, the protective effects of H_2_S at the vascular level are related to inhibition of endothelial phenotype maintenance [109].

## 7. Conclusions

In the cardiovascular system, fibrosis proceeds through multiple steps by way of tissue reparation after damage, ultimately resulting in excessive deposition of connective tissue that alters the morphology and functionality of vessels and the heart. The pivotal role of the EndMT process is emerging in these pathological changes, and therefore therapeutic interventions focused to containment or inversion of the transition from endothelial to mesenchymal phenotype are an interesting area of intense research today. In this context, H_2_S and molecules able to release H_2_S have recently been associated with anti-fibrotic properties in the cardiovascular system. Despite the limited evidence in animal and human studies, several pathways recognized as part of EndMT (TGF-β) seem to be involved in the protective effects of this gasotransmitter, thus suggesting that H_2_S could be a relevant physiological modulator of the fibrotic process. Of course, this review could open a discussion about the need for further studies confirming published evidence and aiming to better define the role of H_2_S in the EndMT/fibrosis process. Indeed, taking into account the different diseases associated with a negative regulation of H_2_S levels, molecules able to release H_2_S could represent a useful tool to slow down the progression of cardiovascular fibrosis as well as a possible approach to pursue in pharmacological therapy. 

## Figures and Tables

**Figure 1 antioxidants-10-00910-f001:**
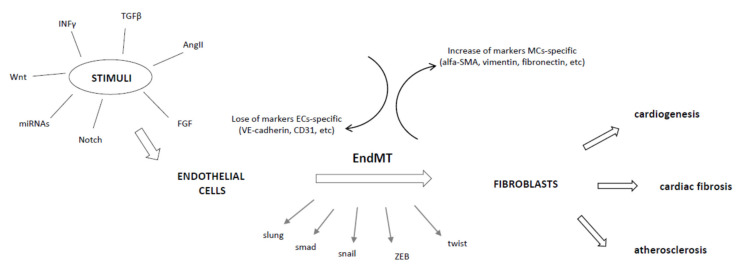
Schematic representation of molecular mechanisms underlying the EndMT phenomenon.

**Figure 2 antioxidants-10-00910-f002:**
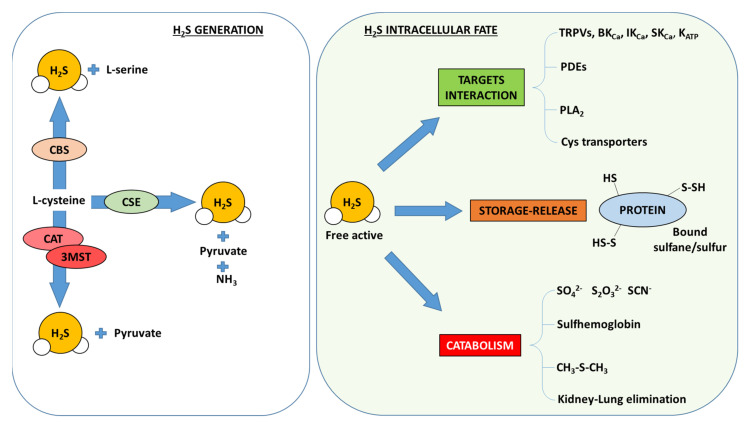
Biosynthesis and catabolism of H_2_S.

**Figure 3 antioxidants-10-00910-f003:**
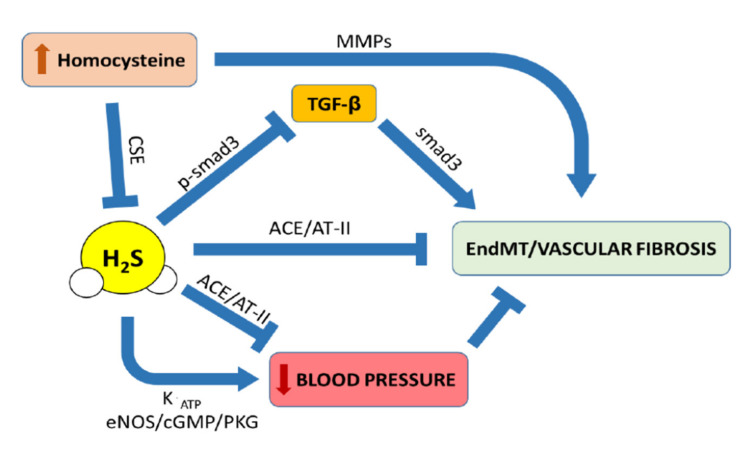
Multiple pathways associated with the effects of H_2_S against vascular fibrosis.

**Figure 4 antioxidants-10-00910-f004:**
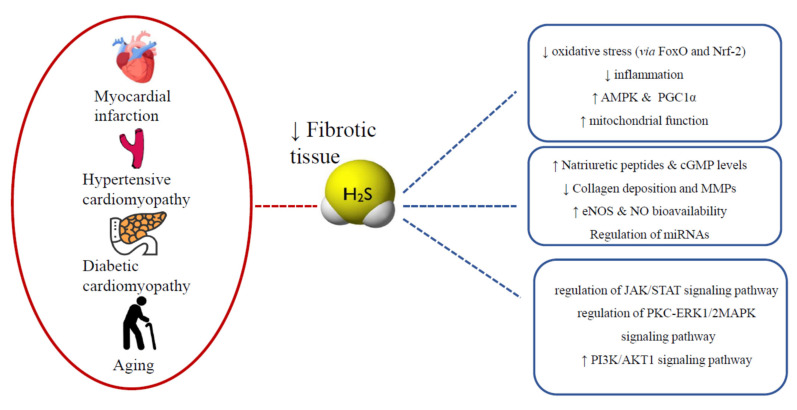
Multitarget effects of H_2_S against myocardial fibrosis.

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
