# Peer review of "Modulation of EndMT by Hydrogen Sulfide in the Prevention of Cardiovascular Fibrosis"

_antioxidants, 2021, doi:10.3390/antiox10060910_

Round 1

Reviewer 1 Report

This is a very nice article focused on the hydrogen sulfide and endothelial mesenchymal transition (EndMT). I have the following suggestions for the authors.

(1) The flow of the article could be improved by reducing the number of paragraphs. It is hard to tell if this was intentional or due to formatting.

(2) The authors provide evidence that hydrogen sulfide therapy can target many of the signaling events leading to the development of fibrosis. In this context, the authors suggest that hydrogen sulfide therapy could be beneficial in the reduction of EndMT. However, there is very little evidence for this. The authors should highlight the limited work on the subject and provide some insight for future studies.

(3) The authors should highlight the evidence for a decrease in hydrogen sulfide levels (pathology induced, not genetic) in instances where there is a propensity for EndMT.

Author Response

This is a very nice article focused on the hydrogen sulfide and endothelial mesenchymal transition (EndMT). I have the following suggestions for the authors.

REPLY: Thank for the nice comment.

(1) The flow of the article could be improved by reducing the number of paragraphs. It is hard to tell if this was intentional or due to formatting.

REPLY: We revised the text, in order to improve the flow of the article.

(2) The authors provide evidence that hydrogen sulfide therapy can target many of the signaling events leading to the development of fibrosis. In this context, the authors suggest that hydrogen sulfide therapy could be beneficial in the reduction of EndMT. However, there is very little evidence for this. The authors should highlight the limited work on the subject and provide some insight for future studies.

REPLY: Thank for your suggestion, we have stressed it in the conclusions.

(3) The authors should highlight the evidence for a decrease in hydrogen sulfide levels (pathology induced, not genetic) in instances where there is a propensity for EndMT.

REPLY: We inserted this aspect, thank you for suggestion.

Reviewer 2 Report

This is a review of focused on the contribution of endothelial mesenchymal transition and how it relates to hydrogen sulfide-mediated prevention of fibrosis in the cardiovascular system. There are many concerns that should be addressed before the review is ready for publication.

  1. There are extensive corrections needed to sentences throughout the document. One example is "...in which mesenchymal phenotype can recovery endothelial feature..."
  2. There are multiple one sentence paragraphs.
  3. Section 2 is very disorganized. One possibility is to model this section after figure 1 and split it into defined signals that influences EndMT.
  4. Sections 5 and 6 do not discuss EndMT at all and this would be a great opportunity to thread in this relationship

  5. Section 6 talks about the role of hydrogen sulfide in EndMT however it doesnt adequately satisfy the title of this review. This section describes more the relationship of EndMT and hydrogen sulfide in cell culture which would not adequately recapitulate physiological changes as cell culture induces EndMT on its own.

  6.  

Author Response

This is a review of focused on the contribution of endothelial mesenchymal transition and how it relates to hydrogen sulfide-mediated prevention of fibrosis in the cardiovascular system. There are many concerns that should be addressed before the review is ready for publication.

  1. There are extensive corrections needed to sentences throughout the document. One example is "...in which mesenchymal phenotype can recovery endothelial feature...
  2. There are multiple one sentence paragraphs.

REPLY: We deeply revised the manuscript as regard the language and the organization of the paragraphs.

  1. Section 2 is very disorganized. One possibility is to model this section after figure 1 and split it into defined signals that influences EndMT.

REPLY: thank for your suggestion, the position of Figure 1 has been established by editoral staff, we re-organized the section in order to obtain a clearer section.

  1. Sections 5 and 6 do not discuss EndMT at all and this would be a great opportunity to thread in this relationship

REPLY: we revised the section 5 and 6 to highlight focus on EndMT.

  1. Section 6 talks about the role of hydrogen sulfide in EndMT however it doesnt adequately satisfy the title of this review. This section describes more the relationship of EndMT and hydrogen sulfide in cell culture which would not adequately recapitulate physiological changes as cell culture induces EndMT on its own.

REPLY: We discussed this aspect in the conclusions, moreover we proposed a new title.

Reviewer 3 Report

In Testai et al., "Contribution of EndMT in the hydrogen sulfide-mediated prevention of fibrosis in cardiovascular system", the authors describe the key role of endothelial-mesenchymal transition in pathogenesis of such cardiovascular diseases as atherosclerosis, hypertension, diabetes, and heart failure. They go on to explain the signaling underlying endothelial-mesenchymal transition, leading to fibrosis. The authors then introduce H2S, as a naturally produced autocrine/paracrine agent which protects against endothelial-mesenchymal transition and fibrosis.

The review is comprehensive and very interesting. It is an excellent addition to the special issue of Antioxidants and will no doubt be of interest to cardiovascular researchers across a wide range of specialties.

In this reviewers opinion, the primary downfall of the review is its complexity, a reflection of the complexity of the topic. This reviewer found that the concepts were clearly described in each paragraph, but that the overall structure of the review made the information difficult to comprehend as an integrative whole. This does not appear due to a lack of understanding or mastery over the topic by the authors. Instead, in this reviewer's opinion, the confusion and difficulty understanding the 'big picture' or 'key messages' is due to the organization. Please refer to the 'major suggestions' below.

There are also a number of grammatical and spelling mistakes. This reviewer would like to congratulate the authors for writing an eloquent review in what is presumably a second language. However, a number of small changes can be made to correct the English and improve the manuscript. Please refer to the 'minor suggestions' below.

Major Suggestions:

  1. May this reviewer suggest that the authors reconsider the organization of the review? It may require more subheadings to help the reader follow the wealth of information. It may also help to have additional thematic sections, such as integrating information by disease or by animal model.
  2. In this reviewer's opinion, a list of abbreviations would be very helpful.
  3. The review includes information on lung cancer and peritoneal fibrosis. This reviewer was unsure that these elements contributed to the review, as the focus was on cardiovascular diseases. The link/importance of this information may need to be clarified.
  4. This reviewer found that, with so much complex information, the authors' grammatical organization sometimes suffered. Paragraphs containing a single sentence were present throughout the manuscript. A paragraph must contain a minimum of three sentences, as per https://courses.lumenlearning.com/: "Every paragraph in the body of an essay consists of three main parts: a topic sentence, some supporting sentences, and a concluding sentence". It is this reviewer's belief that many of the sentences presented incorrectly as paragraphs can be integrated into other parts of the text, where they are thematically linked. Alternatively, where a sentence is presented as a paragraph, the authors may wish to expand on the information to create a full paragraph.

Minor Suggestions:

  1. In this reviewer's understanding, key words should not repeat words in the title, as search engines look at both. Suggest removing 'hydrogen sulfide' and 'fibrosis' from the key words and replacing with novel terms such as 'canonical and non-canonical smad signaling' or the diseases of particular interest in the paper.
  2. Please be more explicit at the sources of your information. For example, paragraph 1 of the introduction (lines 28 to 38) lacks any references.
  3. Where the authors write 'Noteworthy, ...", the more correct wording would be 'Of note, ..."
  4. The abbreviations NO, TRPV, GSH, PPAR, GSSG, cGMP-PKG, ACE, I/R, ANP, BNP have not been explained at their first use. In other cases, abbreviations are not consistently used. For example, H2S, HHcy and MitoKATP/MitoKATP.
  5. Some misspelled words are present. For example, embriogenesis, occurr, trasctiption, phosphorilation/lates, coltured.
  6. In line 139, the authors write that a point was 'proven'. Please note that in scientific method, information may not be proven unless as a grand theory. This reviewer suggests instead writing that "Illigens and colleagues showed...".
  7. Should abbreviations be spelled out in figure titles?

Author Response

In Testai et al., "Contribution of EndMT in the hydrogen sulfide-mediated prevention of fibrosis in cardiovascular system", the authors describe the key role of endothelial-mesenchymal transition in pathogenesis of such cardiovascular diseases as atherosclerosis, hypertension, diabetes, and heart failure. They go on to explain the signaling underlying endothelial-mesenchymal transition, leading to fibrosis. The authors then introduce H2S, as a naturally produced autocrine/paracrine agent which protects against endothelial-mesenchymal transition and fibrosis. The review is comprehensive and very interesting. It is an excellent addition to the special issue of Antioxidants and will no doubt be of interest to cardiovascular researchers across a wide range of specialties.

REPLY: Thank you very much, we are gratefull for your comments.

In this reviewers opinion, the primary downfall of the review is its complexity, a reflection of the complexity of the topic. This reviewer found that the concepts were clearly described in each paragraph, but that the overall structure of the review made the information difficult to comprehend as an integrative whole. This does not appear due to a lack of understanding or mastery over the topic by the authors. Instead, in this reviewer's opinion, the confusion and difficulty understanding the 'big picture' or 'key messages' is due to the organization. Please refer to the 'major suggestions' below. There are also a number of grammatical and spelling mistakes. This reviewer would like to congratulate the authors for writing an eloquent review in what is presumably a second language. However, a number of small changes can be made to correct the English and improve the manuscript. Please refer to the 'minor suggestions' below.

REPLY: Thank for you suggestion, we deeply revised the manuscript, in order to correct the language and ameliorate the flow of the text.

Major Suggestions:

  1. May this reviewer suggest that the authors reconsider the organization of the review? It may require more subheadings to help the reader follow the wealth of information. It may also help to have additional thematic sections, such as integrating information by disease or by animal model.

REPLY: Really we considered re-organising the manuscript by inserting sub-headings (for paragraph 2, we did), but in some cases this was not possible, we believe that this could over-fragment the text.

  1. In this reviewer's opinion, a list of abbreviations would be very helpful.

REPLY: We inserted it. 

  1. The review includes information on lung cancer and peritoneal fibrosis. This reviewer was unsure that these elements contributed to the review, as the focus was on cardiovascular diseases. The link/importance of this information may need to be clarified.

REPLY: We reported information on lung cancer and peritoneal fibrosis because they are, at the moment, the unique studies in which H2S has been correlated to EndMT process, in this regard we stressed the lack of knowledge in the conclusion section.

  1. This reviewer found that, with so much complex information, the authors' grammatical organization sometimes suffered. Paragraphs containing a single sentence were present throughout the manuscript. A paragraph must contain a minimum of three sentences, as per https://courses.lumenlearning.com/: "Every paragraph in the body of an essay consists of three main parts: a topic sentence, some supporting sentences, and a concluding sentence". It is this reviewer's belief that many of the sentences presented incorrectly as paragraphs can be integrated into other parts of the text, where they are thematically linked. Alternatively, where a sentence is presented as a paragraph, the authors may wish to expand on the information to create a full paragraph.

REPLY: we revised the manuscript.

Minor Suggestions:

  1. In this reviewer's understanding, key words should not repeat words in the title, as search engines look at both. Suggest removing 'hydrogen sulfide' and 'fibrosis' from the key words and replacing with novel terms such as 'canonical and non-canonical smad signaling' or the diseases of particular interest in the paper.
  2. Please be more explicit at the sources of your information. For example, paragraph 1 of the introduction (lines 28 to 38) lacks any references.
  3. Where the authors write 'Noteworthy, ...", the more correct wording would be 'Of note, ..."
  4. The abbreviations NO, TRPV, GSH, PPAR, GSSG, cGMP-PKG, ACE, I/R, ANP, BNP have not been explained at their first use. In other cases, abbreviations are not consistently used. For example, H2S, HHcy and MitoKATP/MitoKATP.
  5. Some misspelled words are present. For example, embriogenesis, occurr, trasctiption, phosphorilation/lates, coltured.
  6. In line 139, the authors write that a point was 'proven'. Please note that in scientific method, information may not be proven unless as a grand theory. This reviewer suggests instead writing that "Illigens and colleagues showed...".
  7. Should abbreviations be spelled out in figure titles?

REPLY: Thank you, we changed the text following your suggestions.

Round 2

Reviewer 2 Report

This manuscript has been improved but still needs editing.

  1. Line 40 is one sentence and should not be a paragraph.
  2. Line 78 is one sentence and should not be a paragraph.
  3. Line 69 is one sentence and should not be a paragraph it has been unchanged from the first submission.
  4. Line 114 is one sentence and should not be a paragraph.
  5. This is not a comprehensive list.

Author Response

Dear referee,

we revised the manuscript and in yellow color we highlighted the changes. Moreover, during the revision, we observed that, probably during the formatting in the template, a paragraph was lacking (par. 3.1); then we added it.

-Line 40 is one sentence and should not be a paragraph.

-Line 78 is one sentence and should not be a paragraph.

-Line 69 is one sentence and should not be a paragraph it has been unchanged from the first submission.

-Line 114 is one sentence and should not be a paragraph.

REPLY: we completed the revision on the basis of your suggestions as regard the organization of the text.

This is not a comprehensive list.

REPLY:  we attached an abbraviation list, we apologize for the forgetfulness.

Round 3

Reviewer 2 Report

All concerns have been addressed.